# Phyx.io: Expert-Based Decision Making for the Selection of At-Home Rehabilitation Solutions for Active and Healthy Aging

**DOI:** 10.3390/ijerph19095490

**Published:** 2022-05-01

**Authors:** Javier Dorado Chaparro, Jesús Fernández-Bermejo Ruiz, María José Santofimia Romero, Xavier del Toro García, Rubén Cantarero Navarro, Cristina Bolaños Peño, Henry Llumiguano Solano, Félix Jesús Villanueva Molina, Anabela Gonçalves Silva, Juan Carlos López

**Affiliations:** 1Computer Architecture and Networks Group, School of Computer Science, University of Castilla-La Mancha, 13071 Ciudad Real, Spain; jesus.fruiz@uclm.es (J.F.-B.R.); mariajose.santofimia@uclm.es (M.J.S.R.); xavier.deltoro@uclm.es (X.d.T.G.); ruben.cantarero@uclm.es (R.C.N.); cristina.bolanos@uclm.es (C.B.P.); henry.llumiguano@uclm.es (H.L.S.); felix.villanueva@uclm.es (F.J.V.M.); juancarlos.lopez@uclm.es (J.C.L.); 2Center for Health Technology and Services Research, Health Sciences School, University of Aveiro, 3810-193 Aveiro, Portugal; asilva@ua.pt

**Keywords:** healthy and active ageing, video-based system, at-home rehabilitation

## Abstract

While the importance of physical activity in older adults is beyond doubt, there are significant barriers limiting the access of older adults to physical exercise. Existing technologies to support physical activity in older adults show that, despite their positive impacts on health and well-being, there is in general a lack of engagement due to the existing reluctance to the use of technology. Usefulness and usability are two major factors for user acceptance along with others, such as cost, privacy, equipment and maintenance requirements, support, etc. Nevertheless, the extent to which each factor impacts user acceptance remains unclear. Furthermore, other stakeholders, besides the end users, should be considered in the decision-making process to develop such technologies, including caregivers, therapists and technology providers. In this paper, and in the context of physical rehabilitation and exercise at home, four different alternatives with incremental characteristics have been defined and considered: a software-based platform for physical rehabilitation and exercise (Alternative 1), the same software platform with a conventional RGB camera and no exercise supervision (Alternative 2), the same software platform with a convention RGB camera and exercise supervision (Alternative 3) and finally, the same software platform with a depth camera and exercise supervision (Alternative 4). A multiple attribute decision-making methodology, based on the ordinal priority approach (OPA) method, is then applied using a group of experts, including end users, therapists and developers to rank the best alternative. The attributes considered in this method have been usefulness, cost, ease of use, ease of technical development, ease of maintenance and privacy, concluding that Alternative 3 has been ranked as the most appropriate.

## 1. Introduction

The Green Paper on Ageing [1], published by the European Commission, analyzes the main challenges posed by the progressive ageing of the population and the implications this has on economic growth, fiscal sustainability, health and long-term care, well-being, and social cohesion. New approaches, such as those based on the digital transition of services, are being called for to face such challenges.

One of the services whose digital transition could have a great impact on health and well-being is that of physical activity. In fact, according to the WHO [2] “at least 80% of all heart disease, stroke and diabetes and 40% of cancer could be prevented” by tackling the most common risk factors underlying the most prevalent chronic conditions, such as unhealthy diets, physical inactivity, hypertension, and obesity. The importance of physical activity in older adults is not under question [3]; however, it can be very challenging as, in their majority, they already suffer from a condition that can worsen or have a negative effect when there is no expert supervising the exercise performance [4,5]. Nonetheless, attending to supervised and person-centered sessions is not always possible for several reasons, such as economic cost, time constraints, the impossibility to travel to a health center on a daily basis, or simply the lack of such a service. Having access to physical activity programs, as in the case of those individuals living in long-term care facilities, is not a guarantee of engagement in them [6]. According to [7], only 10% of older adults residing in long-term care facilities engage in physical activities twice a week.

The major barriers found by older adults when engaging in exercising are, among the most relevant ones, a lack of time [8], a lack of company [9], a lack of an understanding of the importance of physical activity [10], physical problems, a lack of accessibility, or a fear of falling [11]. Digital solutions that specifically tackle such barriers would have a greater chance of succeeding in engaging older adults in exercising.

Limitations on outdoor mobility are some of the first limitations to occur [12] as people age. The need to exercise from home was also evident during the COVID-19 pandemic. Either because of the mobility restrictions or as a preventive measure, many older adults that periodically attended physical exercise classes saw their activity truncated. Despite the efforts of public authorities to promote physical activity during the lockdown, the authors in [13] concluded that individuals over 55 years old reported a reduction in exercise performance [14,15]. In this sense, the pandemic has made evident not only the need for support systems for at-home physical exercise, but also the need to reach those older adults that have very limited mobility outdoors. The authors in [13] carried out a systematic review of home-based exercise programs designed for populations over 65 with no minimal supervision. The reviewed works were compared on the basis of health or skill-related physical fitness (e.g., strength, muscle power, and balance). The study concluded that home-based exercise seems to improve the aforementioned indicators in healthy older adults over 65. The use of virtual reality can also be found in the literature for purposes related to physical activity, such as balance rehabilitation [16] or gait rehabilitation [17], or more generally to improve physical, mental, or psychosocial health outcomes [18]. Other approaches report the use of wearables, such as accelerometers, or self-reported assessments [19].

From a review of the state of the art, it can be concluded that different approaches can be implemented to support physical activity at home. These approaches have different advantages and disadvantages, depending on the optics from which they are evaluated. For example, the cost of a system can be a determinant factor for end users, who have to pay for it, but not that important for specialists of long-term care facilities, as they will be able to save costs while providing access to exercise support and monitor more people. Similarly, having access to information such as joint ranges or estimated muscle strength might not be relevant for end users, who might not know how to interpret this information, but will be very important for specialists.

This decision-making problem has been stated in the context of the SHAPES Project. The SHAPES project (https://shapes2020.eu/, accessed on 20 March 2022) (Smart and Healthy Ageing through People Engaging in Supportive Systems) is a European-funded Innovation Action intended to promote long-term healthy and active ageing as well as to maintain a high-quality standard of life. In order to do so, SHAPES provides a set of digital solutions, deployed on an EU-standardized open platform, supporting the factors that determine a healthy and active ageing. The different stakeholders involved in the SHAPES Project therefore faced the decision of selecting an approach that will satisfy, in a more comprehensive manner, the interests and desires of those involved in the matter. This common type of problem has been traditionally addressed from the theory of multi-criteria decision-making (MCDM).

More specifically, the SHAPES project has considered four different alternatives to support physical rehabilitation for older adults (through exercises) at home. These alternatives will be evaluated and ranked based on multiple attributes, encompassing aspects such as those related to the aforementioned barriers as wellas others related to technology such as usability, acceptance, or willingness to use. The decision about what system should be employed to evaluate the impact of the intervention has to be previously based on an analysis of the different dimensions involved in this matter. It is not enough to base the decision on the sole criterion of accuracy, acceptance, or engagement. Moreover, different individuals should be considered in this decision—not only the end users. Physicians or therapists, caregivers, end users, and developers should all participate in the decision-making process. As stated by [20] scholars are increasingly resorting to the theory of multi-criteria decision-making (MCDM) to facilitate complex decision-making procedures. MCDM encompasses two categories of methods: multiple-objective decision-making (MODM) models and multiple attribute decision-making (MADM) [21]. The main difference between these two categories is that MODM addresses a continuous space, whereas MADM methods address a discrete one. Among the different problems faced by MCDM methods, the choice problem is one of them [22]. The problem faced here will be addressed as a problem of stating the ordinal priority of different alternatives, according to the approach proposed in [20].

The main objective of this research is therefore the proposal of a multiple attribute decision-making problem, based on the ordinal priority approach (OPA) method, whose solution ensures the technological acceptance, success, and feasibility of the proposed solution. This paper describes the mathematical model and the optimization method applied [20] to support the decision of the most appropriate approach for supporting physical activity in older adults.

## 2. Materials and Methods

### 2.1. Design

The OPA method for multiple attribute group decision-making [20] was chosen to support the selection of the most appropriate alternative for supporting physical rehabilitation for older adults at home. Four alternatives, three typed of individuals, and a set of six attributes are simultaneously considered in the decision-making process.

The main advantage of this OPA method for MCDM is that it simplifies the process of selecting the most appropriate alternative, among different ones that are characterized by different attributes and in which different experts are involved. These are the three edges of the decision-making triangle (i.e., experts, attributes, and alternatives). The work in [20] defines the following concepts:

**Definition** **1.**
*A set of experts is the set I of individuals or decision-makers that have been considered for the evaluation of the different attributes. In this context, the following stakeholders have been considered: developers, physical therapists or physicians in general, caregivers, and older adults. I is formally stated as the set of individual experts $\forall i_{i}\in I$, i being the index of preference of all the experts with i∈(1,…,p).*


**Definition** **2.**
*A set of attributes is the set J of indicators that might be categorized, which will be used to evaluate the different alternatives from different dimensions. J is formally stated as the set of individual experts $\forall j_{j}\in J$, j being the index of preference of all attributes with j∈(1,…,n).*


**Definition** **3.**
*A set of alternatives is the set K of different systems under evaluation. Four major approaches have been considered according to the video source employed. The first alternative is a system that does not consider a video input and therefore does not supervise, in any manner, how the user performs the exercise. The second alternative employs video but just to support video calls, as no supervision of exercises is provided. Finally, there are two alternatives considering two different types of video images: one with depth information and one with RGB video. Among the different depth cameras, there are a long list of possible options. K is formally stated as the set of individual experts $\forall k_{k}\in K$, k being the index of all possible alternatives k∈(1,…,m).*


**Definition** **4.***The objective function Z is the mathematical model to be optimized based on the method proposed in [20]. This function has been defined in* [20] *as follows:*
(1)$$Z\leq i(j\left(r\left(W_{ijk}^{r}-W_{ijk}^{r+1}\right)\right))\;\forall i,j,k,\;\mathrm{and}\;r$$
(2)$$Z\leq ijmW_{ijk}^{m}\;\forall i,j,\;\mathrm{and}\;k$$
*Thus,*

(3)
$$\sum_{i=1}^{p}\sum_{j=1}^{n}\sum_{k=1}^{m}W_{ijk}=1$$


(4)
$$W_{ijk}\geq 0\;\forall i,j,\;\mathrm{and}\;k$$


*$W_{ijk}^{r}$ is the weight of the kth alternative for the jth attribute provided by the ith expert at the rth rank. For each attribute and for each expert, the different alternatives can be ranked accordingly:*

(5)
$$A_{ijk}^{1}\geq A_{ijk}^{2}\geq \ldots \geq A_{ijk}^{r}\geq A_{ijk}^{r+1}\geq \ldots \geq A_{ijk}^{m}\;\forall i,j,k$$



### 2.2. Participants

The decision-making problem addressed in this study is framed in an endeavour to provide a system for at-home rehabilitation under the Pilot Theme 6 (https://shapes2020.eu/about-shapes/pilots/, accessed on 20 March 2022). Therefore, the group of participants in this study was composed of nine experts representing the different stakeholders of this Pilot Theme and were aged between 30 and 68 years old (mean age = 45 years old).

The developers in charge of leveraging such digital solutions were one set of stakeholders. SHAPES projects have proposed a co-creation, co-design, and co-development methodology under which a continuous interaction among technology developers and end users is expected. Developers have to design and implement solutions that answer user needs, so their opinion is also relevant in the decision-making process. Three developer experts with an average age of 40 years old participated in the study.

Specialists supervising the rehabilitation process or the general well being of an individual were considered as another set of experts that should be involved in the decision-making process. This set is comprised of physical therapists, occupational therapists, nurses, and general practitioners or physicians. Although their skills and targeted interventions are different, they were considered part of the same set. Three therapist or physiotherapist experts with an average age of 35 years old participated in the study.

The last set of experts considered in this decision-making process was comprised of different types of end users, including those exercising at home or living in a long-term care facility. This set also included informal caregivers who might eventually decide to invest in such a solution, such as the son that decides to pay for such a system because he is aware of the potential positive benefits that physical activity has on ageing. For the case of caregivers, this does not necessarily imply that the older adult is not capable of making the decision by him/herself (due to a cognitive impairment, for example), although this might be the case. Three end-user or caregiver experts with an average age of 61 years old participated in the study.

In Section 2.4.2, the information of each of the experts who participated in the study is detailed, according to the role, professional position, experience, and educational level. This information will be used to rank the experts who participated in the study, and it will thus be possible to determine the weight of their decisions.

The recruitment process was undertaken under the participants of Pilot Theme 6 and the partners of the SHAPES Project. Informed consent was collected.

### 2.3. Alternatives

This section reviews the different alternatives that are considered in the decision-making process. The different alternatives will involve different hardware and software components. Four alternatives have been considered as digital solutions that support at-home physical rehabilitation:Alternative 1: A digital solution for physical rehabilitation that does not provide video-based supervision/monitoring support.Alternative 2: A digital solution that is equipped with a camera but without any support for posture estimation or, in other words, without support through the supervision/monitoring of exercise performance. The camera is only used for video recording and video call functions.Alternative 3: A digital solution with a camera and software that provides information on the estimation of the body posture and therefore supervises or monitors the performance of exercises.Alternative 4: A digital solution using a special type of camera with depth information, used for a more accurate estimation of body position.

The different experts considered in this study might not hold sufficient knowledge to comment on a specific attribute of a given alternative. This, for example, might be the case when the attributes regard the elements employed to build the system, such as the hardware or software required for the different alternatives. For this reason, aspects regarding the hardware and software elements involved in the different alternatives are detailed below. They were summarized to the experts questioned so that they could rank the attributes of the different alternatives.

The following subsections provide further details of the different features of the considered alternatives. Because Alternatives 1 and 2 are direct, as the first one does not involve any video and the second one only uses it for video-call functionalities, they have been left aside and the focus is on Alternatives 3 and 4.

#### 2.3.1. Alternative 3

This alternative consists in the use of 2D video from which depth information can be calculated using different machine learning techniques. This alternative is therefore intended to estimate body pose from the 2D information retrieved from the RGB video. This alternative is mainly characterized by the availability and low cost of convectional RGB cameras. Table 1 summarizes some examples complying with the required features for the purpose of body pose estimation, as it has a high resolution, a high FPS (frames per second), and a high transference rate, so that these images can be processed in real time.

Machine learning techniques can be used to obtain depth information from 2D sources. In this sense, there are three different type of techniques [23]: supervised, semi-supervised, and unsupervised/reinforced learning. Furthermore, there is a specific type of unsupervised learning, known as deep learning, that involves a neural network with more than one hidden layer. This neural network is capable of extracting properties from input data that can later on be used to classify such data. These neural networks will tune their attributes, or in other words learn, by training. For the training stage, it will be necessary to introduce a large volume of data. For example, the training for human-body pose estimation will involve providing person images. When designing the neural network, it is possible to define the number of properties to be extracted and how to extract them. In this way, the processing power required for the task is fully controlled, as well as the limits of the learning process: the illumination of the environment, the distance of the object being detected by the camera, etc. The neural network design is not an easy process. On the contrary, it requires substantial knowledge in the field of computer vision. Luckily, nowadays, there are tools and libraries that, besides creating and training neural networks, also enable the learned data to be extracted as a file so that it can be used in a “static” way (without being able to modify its behaviour). These files are known as models. There are a great variety of libraries for this purpose, and some even provide models than can be directly employed. Two of the most well known are [24] TensorFlow (https://www.tensorflow.org/, accessed on 15 March 2022) and PyTorch (https://pytorch.org/, accessed on 15 March 2022).

Human-body pose estimation, whether involving motion or not, is one of the main challenges addressed by computer vision, and one on which progress continues to be made [25]. Estimating a human-body pose, in 2D, from a monocular image is a task that can now be successfully achieved. The challenge is found in performing such estimation in 3D, because of the great ambiguity in the 2D positions of human-body joints estimated from a 2D image [25]. An additional limitation is the computational overhead that is required to calculate these 3D positions, which undeniably impacts the system performance.

There are different alternatives that can be employed to estimate 3D body pose from monocular images. One of the alternatives consists in the use of the MoveNet Singlepose model [26] based on the architecture MoveNetV2 [27]. This model has been designed to process RGB images and retrieve from them a list of joints with their positions in 2D coordinates. This model is available for download from the TensorFlow Hub [26]. This model has been optimized for the quick detection of joints, as it is thought to be employed in the recognition of fitness activities. Nonetheless, the performance in embedded systems can be compromised, as the model was originally designed for conventional computing devices.

Alternatively, MediaPipe Pose [28] can also be employed for the purpose of pose estimation. In this case, MediaPipe employs two different models, in a sequential manner, as described below:BlazePose Detector: This model is intended to process RGB images and yield one of the following results:the position of the center of the detected hips;the radius of the circumference surrounding the detected person;the inclination angle of the straight line connecting the center of the hips to the center of the shoulders.BlazePose GHUM 3D [29]: This model processes both the image and the regions of Interest (ROIs) detected from the previous model, calculating the position of 33 joints that, after a post-processing stage, yields two elements:Pose Landmarks: These contain the list of the position, in 3D coordinates, for the 33 joints. The depth information of every joint (i.e., the z coordinate) has its coordinate origin in the hip center, previously calculated. The depth information will increase as the person moves away the camera. Additionally, a percentage of the visibility of every joint is also provided.Pose World Landmarks: These contain the same information as the Pose Landmarks, although the values are provided in meters rather than in pixels, as for the previous one.

Both models have different versions whereby each can offer different results in terms of accuracy and/or processing speed. These versions are compared in the aforementioned terms in Table 2 and Table 3. Other aspects that affect TensorFlow model performance are its format [30] and the type of data it processes. Both depend on how the model has been designed and whether it has been optimized in any way [30].

It is worth mentioning that none of the models, when performing inference or processing, stores and/or sends any information about the image or the detected person. Therefore, the privacy of the user is preserved.

#### 2.3.2. Alternative 4

This alternative proposes the use of depth information to inform about the position of the different body joints while performing exercise routines. The main asset of this alternative is the precision of the depth information retrieved using depth sensors, rather than performing an estimation from the 2D information, as in the previously discussed alternative.

Depth cameras, or cameras equipped with depth sensors, are used to obtain depth readings in 3D environments. This information turns out to be useful in applications such as industrial environments [31,32], robotics [33,34], autonomous vehicles [35,36], security [37,38], and even medicine [39,40,41]. Skeletal tracking is one of the most widespread applications. Due to its high accuracy and the advances in the field of machine and deep learning, it is possible to identify in real time a wide variety of joints in the human body. Recent studies have demonstrated the usefulness of such information in the field of sports [42] for obtaining metrics regarding the athlete’s performance, identifying postures to improve, or correcting movements to avoid possible injuries [43]. Another line of research in which this type of information has generated great expectations has been in the area of human action recognition (HAR), giving rise to a wide variety of works [44,45,46,47]. Much work has also been done in the field of rehabilitation, as tracking of the human skeleton makes it possible to help guide and monitor rehabilitation activities [48,49,50,51,52].

Depth information can be collected using different approaches, as discussed below:Stereo Vision: This approach closely resembles the way humans see the world, as it uses two cameras to mimic how human eyes collect depth information from the environment.Structured Light: This method requires projecting a certain light pattern by means of a projector, while one or more cameras (placed at a known angle with respect to the projector) are in charge of identifying this pattern in the environment. To obtain the depth information, the difference between the projected pattern and the distorted pattern captured by the cameras is calculated.Time Of Flight (ToF): In this approach, light patterns are emitted by lasers in the infrared spectrum and their reflection is captured by a receiver. To obtain the depth map, the Time of Flight (ToF) concept is used. This is the time it takes for light to reach a certain distance. Because the speed of light is known, it is possible to determine the distance of objects by interpreting the time it takes for the projected infrared light to come and go. The time required is directly proportional to the distance.

Besides the employed depth technology, shown in Table 4, there are other technical parameters that have to be considered, such as the effective depth range, the possible resolution settings, the number of frames per second at which the device can operate, and the development tools available for developers. Table 5 compares the main devices found on the market, based on these factors.

It is quite common for different manufacturers to offer a software development kit (SDK) along with a device. These development environments have a series of features, tools, and requirements associated with them that must be assessed simultaneously with the technical specifications of the device, as they will be decisive when developing a solution based on this technology. Table 6 provides a comparison based on these factors. Thus, some SDKs natively track the skeleton or the face, while others require the integration of generic third-party tools, such as OpenCV (https://opencv.org/, accessed on 13 March 2022) or Nuitrack (https://nuitrack.com/, accessed on 13 March 2022).

### 2.4. Procedure

The procedure proposed in [20] outlines a set of simple steps that will lead to the ordinal ranking of the considered alternatives, based on the expert opinions about the considered attributes. This procedure aims to determine the most appropriate approach to support physical rehabilitation at home, taking into account different attributes whose trade-offs satisfy the different requirements considered by all stakeholders.

#### 2.4.1. Determining the Attributes

The first step is to determine the attributes that will inform the decision about the most appropriate alternative. Because there are different participants in the decision-making process, there will be different types of attributes. Table 7 summarizes the attributes and sub-attributes considered here, based on the analyst opinion. It has to be noticed that the initial list of attributes was shared with a subset of experts who validated and updated the list.

The analysts compiled a list of attributes that, in their opinion, are relevant for the different stakeholders. The analyst role is here played by the authors of this work, also responsible for conducting the study that will lead to an informed decision about the best alternative for supporting physical rehabilitation at home. The initial list of attributes was validated and updated by different representatives of the expert sets. These experts, one per expert subcategory (developers, medical experts or formal caregivers, and end users including informal caregivers), analyzed the initial list of attributes based on their expertise, and updates were carried out on the list. Table 7 summarizes the agreed list of attributes after several iterations with these experts.

#### 2.4.2. Specifying and Ranking the Experts

Next, the nine experts who participated in the decision-making process are specified. These experts belong to one of the three roles mentioned in Section 2.2. These participants were ranked taking into consideration, in order, their role in the platform. The role with the highest priority was the “end-user/caregiver”, the role with the second-highest priority was the “therapist/physiotherapist”, and the role of “developer” was ranked as having the lowest priority. The ranking also considered which expert has the highest professional position, the number of years of experience, and the level of education. Table 8 summarizes the experts’ characteristics used for the ranking.

#### 2.4.3. Ranking the Attributes

The experts in this step took part in the decision-making process by ranking the attributes listed in the first part of Section 2.4.1. Thus, each expert ranked the importance of these attributes. It was possible that, for some experts, there were attributes that were not relevant to them or they did not know how to rank them. It was also possible that experts rank several attributes equally. In order to rank the attributes, an exact description of each attribute was given to the experts, as shown in Table 7.

#### 2.4.4. Ranking the Alternatives in Each Attribute

Experts were asked to rank each alternative with respect to each of the attributes. In order for the experts to have enough information to do this, as stated in Equation (Equation 6), experts were given the following:Videos explaining the different functionalities of the four alternatives over the Phyx.io Platform (https://youtu.be/sdtMTHLGkFA, accessed on 30 March 2022), the pose estimation video (https://youtu.be/SrbcbDZkhOA, accessed on 30 March 2022), a video showing the performance of an exercise (https://youtu.be/05KisckHgZE, accessed on 30 March 2022) for Alternative 3 and 4, and a video that shows how the video call functionality is accessed using an RFID band (https://youtu.be/SLvdw1IW2fA, accessed on 30 March 2022), which is part of Alternative 2, 3, and 4;a table with the main features and functionalities of the four considered alternatives (Figure 1);tables summarizing the four alternatives, according to each of the considered attributes; Figure 2 shows an example of the usefulness attribute for each of the four alternatives.
(6)$$\left(A_{ijk}^{1},A_{ijk}^{2},\ldots ,A_{ijk}^{m}\right)$$

#### 2.4.5. Solving the Model

Finally, the mathematical model presented in Equations (Equation 1)–(Equation 4) proposed in [20] was applied. Equations (Equation 7)–(Equation 9) were used to obtained the weights achieved by the different alternatives, the considered attributes, and the experts, respectively. Equation (Equation 10) shows how these weights are stated so that the alternatives can be ranked.
(7)$$W_{k}=\sum_{i=1}^{p}\sum_{j=1}^{n}W_{ijk}\;\forall k$$
(8)$$W_{j}=\sum_{i=1}^{p}\sum_{k=1}^{m}W_{ijk}\;\forall j$$
(9)$$W_{i}=\sum_{j=1}^{n}\sum_{k=1}^{m}W_{ijk}\;\forall i$$
(10)$$\left(W_{ijk}^{1},W_{ijk}^{2},\ldots ,W_{ijk}^{m}\right)$$

The upcoming section shows the obtained results yielded from the application of the aforementioned methodology.

## 3. Results

This section presents the results obtained from the application of the methodology proposed in [20] and the involved experts with different backgrounds, as described in Section 2.2. The experts, based on their expertise, contributed their opinions regarding the strengths and weaknesses of the proposed platforms for healthy and active ageing, considering the attributes described in Table 7.

Experts were ranked, as described in Table 8, considering different aspects such as their role, position, experience, and education level. These experts ranked the attributes stated in Table 7 according to their background and expertise. Before ranking the alternatives, information was provided to the experts about the different alternatives.

Table 9 shows the importance ranking granted to different experts. Figure 3 shows which experts have more weight in the decision-making process. The experts with the greatest weight (E7, E8, and E9) are those with an “end user/caregiver” role, and those with less weight in the decision-making coincide are “developers” (E1, E2, and E3).

Table 10 shows that, from the experts’ point of view, the usefulness and ease of use are the most important attributes, and privacy is the attribute they consider least important among the criteria or attributes presented. Figure 4 shows the meaning of each of the criteria considered in the decision-making process.

Furthermore, Figure 5 depicts the attributes that are most important to the experts, showing again that the usefulness (C1) and ease of use (C3) are the most important to almost all experts.

Finally, it can be observed in Table 11 and Figure 6 that the Alternative with the highest rank according to the experts is Alternative 3 (weight = 0.3190) followed by Alternative 1 (weight = 0.2958), with Alternatives 4 (weight = 0.1936) and 2 (weight = 0.1916) closing the ranking with similar weights. The decision making procedure has therefore led to the selection of Alternative 3 as the most convenient approach to provide at-home physical rehabilitation for older individuals.

## 4. Discussion

This study was motivated by the importance of considering the opinions of different experts in the process of deciding what requirements need to be met by a platform designed to support physical activity at home. Previous works [53] have shown that, despite the positive impacts that a certain solution for physical activity might have on users, the fact that the end user perception was not considered at early stages of the software life cycle led to a lack of engagement, therefore failing in its endeavour to promote physical exercise. The study presented is part of a broader objective to eventually provide a system for physical rehabilitation at home. Nevertheless, to ensure user acceptance and therefore to help increase the user’s willingness to use the proposed technological solution, different expert opinions have been considered in the decision-making process. Although end users, being the recipients of the technological solution, are the ones with the highest weight in the decision-making, the opinions of therapists (or health experts in general) along with those responsible for creating the technological solution (i.e., the developers) need to be considered. Because of the weights assigned to the different experts, the obtained rank confirmed that the first three experts were end users, followed by healthcare experts and developers. However, inside the same group of experts, opinions are also weighted based on other criteria, such as role, position, and experience, which might cause that the opinion of two different types of experts might be almost equally important. For example, the weights obtained by Experts E6 and E1 are very similar. In this particular case, the background and expertise of the developer qualifies his/her opinion to contribute to the decision-making process as much as the healthcare expert.

The work in [54] identifies the different barriers found by older adults when using technological solutions for assisted living, which eventually lead to a lack of engagement or an unwillingness to use the proposed solution. These barriers include cognition, physical ability, perception, and motivation. Similarly, the work in [55] identifies the main post-implementation acceptance factors categorized into six themes: (1) concerns such as system malfunctions, false alarms, high cost, stigmatization, and a lack of training; (2) the experienced positive characteristics of the technology such as privacy, increased safety, and unobtrusiveness; (3) the experienced benefits of the technology, such as increased communication, increased capabilities to perform ADL, a reduced burden on family, or a perceived need to use; (4) a willingness to use the technology, which includes the time spent on using (testing) it; (5) social influence, such as the influence of family or the influence of organizations; (6) the characteristics of seniors, such as previous technological experience or their physical environment. These conclusions cohere with the results obtained from the ranking of the considered attributes, where usefulness and ease of use are the most relevant. Similarly, the authors in [56] conclude that users are willing to forfeit privacy in exchange for utility. This same conclusion can be observed in the obtained attribute ranking results, as privacy is considered the least important, and usefulness is considered the most important.

The ease of technical development and maintenance are equally important. These two attributes are mainly relevant for developers, while the ease of use directly affects users and is thus ranked the second most important attribute. Usefulness and ease of use are related, but the fact that usefulness has a higher weight implies that all experts are willing to sacrifice ease of use when usefulness is at stake. In other words, despite the importance of having an easy-to-use system, especially for the older adult population, having more functionalities (which therefore involves complicating the system) is more important. Similarly, privacy, despite the importance that it has for end users, is sacrificed when the end user obtains greater functionality in returns.

The position that system cost takes in the ranking list is also interesting. The fact that usefulness and ease of use are more important than cost means that users are willing to pay more if more useful functionalities are obtained in return. It is also relevant that cost and privacy are ranked very closely. Thus, in terms of acceptance, privacy and cost are almost equally important. Users are willing to pay more to obtain more functionalities, but they are not willing to pay more to obtain higher privacy. This explains why Alternative 4, despite being the most advanced solution, only achieved the third position in the ranking, after Alternative 1, which is the most basic in terms of functionality and the cheapest one.

Alternative 3 had the highest position, meaning that the three sets of experts regarded as the most useful and the easiest to use. The provided functionality, despite not being as advanced as Alternative 4, manages to keep a good trade-off between cost, technological complexity, and maintenance. In terms of privacy, Alternative 3 performs worse than Alternative 1 and 4, as RGB video was employed. Nonetheless, the importance that experts granted to the other attributes managed to compensate such a loss of privacy.

Alternative 3 was closely followed by Alternative 1. The main difference among these two approaches, in comparison with Alternative 2, is the supervision functionality offered by Alternative 3. The importance of supervision or physical exercise monitoring as it pertains to user engagement with physical therapies has been shown in [57,58,59]. The results obtained here confirm the role that receiving feedback and tracking evolution plays in user acceptance, as stated in [48]. The accuracy of the feedback is not considered more important than the price, as can be concluded from the fact that Alternative 4 is ranked similarly to Alternative 2, which does not provide any supervision feedback. The improvement in accuracy resulting from the use of a depth camera does not compensate for the increased complexity and the associated price.

## 5. Conclusions

This paper presents a study carried out to scientifically determine which features should be provided by platforms designed for facilitating physical rehabilitation at home for older adults. There were different stakeholders with different interests, i.e., end users, healthcare professionals, and developers. Furthermore, barriers and facilitators already identified in the literature should be considered to ensure that end users, i.e., those that eventually determine the success of a technological tool for assisted living, accept the platform and are willing to use it.

A MADM method was applied to determine the alternative most favored by the different experts. Four different alternatives for the support of at-home physical rehabilitation were proposed. A thorough analysis was carried out to identify hardware and software that could be used to implement the potential alternatives. This analysis yielded relevant information about the cost and the ease of technical development and maintenance. A group of nine experts in the field of healthy and active ageing, including end users, were considered for the decision-making process.

Alternative 3 obtained the highest rank. This alternative involves video-based monitoring, with feedback and corrections to the user, and employs a low-cost camera. The functionality for feedback and corrections is implemented based on a body pose estimated by the software, rather than using depth information, as proposed in Alternative 4. The precision for pose estimation purposes is superior to that in Alternative 4. The accuracy of the pose estimation based on the software libraries was perceived as useful enough to counteract the lost of privacy, as it also meant that the solution was cheaper.

Future works are currently being undertaken to provide a complete version of a platform for physical rehabilitation at home. End-user acceptance will be also studied along with the impact that the proposed platform has on user health and well being.

## Figures and Tables

**Figure 1 ijerph-19-05490-f001:**
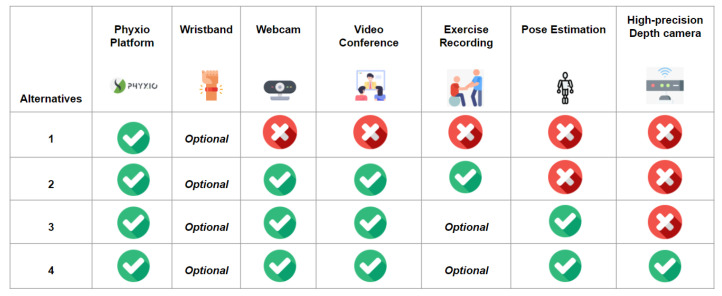
Characteristics of each alternative.

**Figure 2 ijerph-19-05490-f002:**
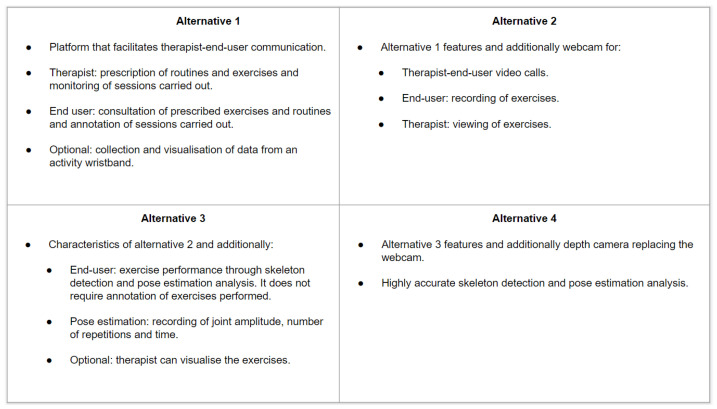
Details of the four alternatives with respect to the usefulness attribute.

**Figure 3 ijerph-19-05490-f003:**
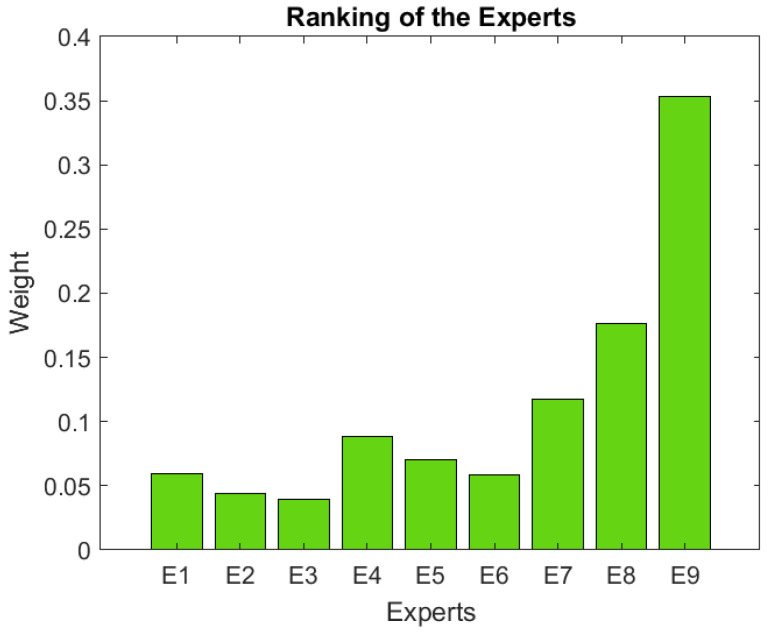
Ranking of the Experts.

**Figure 4 ijerph-19-05490-f004:**
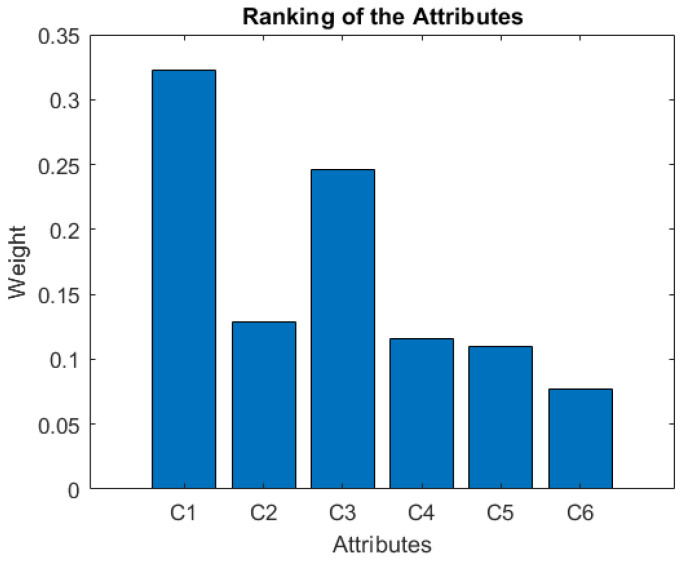
Ranking of the attributes.

**Figure 5 ijerph-19-05490-f005:**
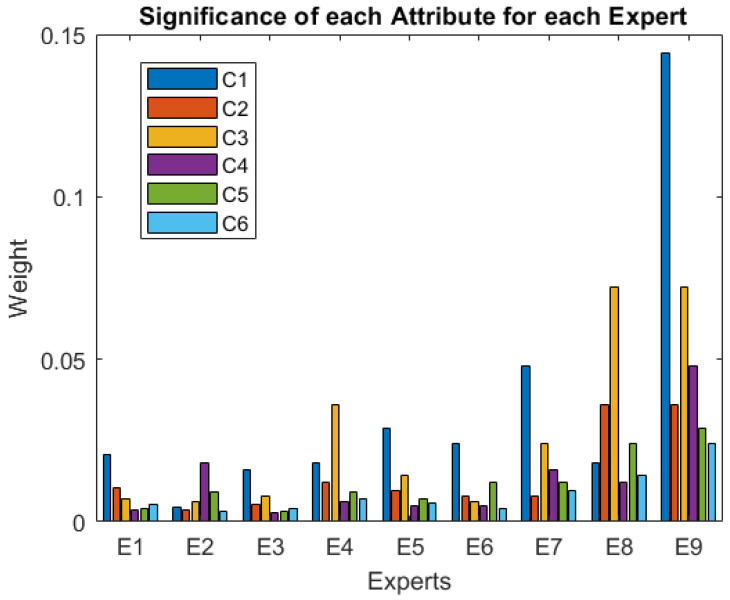
Significance of each attribute for each expert.

**Figure 6 ijerph-19-05490-f006:**
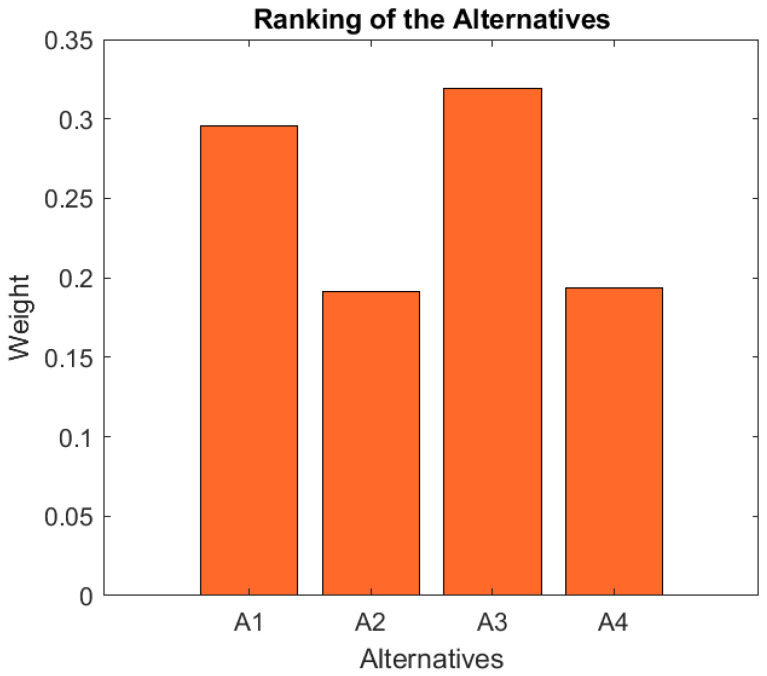
Ranking of the alternatives.

**Table 1 ijerph-19-05490-t001:** Comparison of RGB cameras (prices last updated in March 2022).

	Resolution	FPS	USB	Price (€)
Logitech HD Webcam C270	1280 × 720	30	2.0	34.99
Logitech HD Pro Webcam C920	1920 × 1080	60	3.0	78.97
Logitech HD Webcam C310	1280 × 720	30	2.0	61.99
Owlotech Start Webcam 720p	1280 × 720	-	2.0	29.99
Krom Kam Webcam 1080P HD	1920 × 1080	30	2.0	18.98

**Table 2 ijerph-19-05490-t002:** Characteristics of the pose estimation models considered.

	MoveNetV2	BlazePose GHUM
Joint number	17	33
Is it 3D?	No	Yes
Formats (https://www.tensorflow.org/hub/model_formats, accessed on 17 March 2022)	TF, TFLite, TFJS	TFLite, TFJS
GPU supported?	TF y TFJS	TFJS

**Table 3 ijerph-19-05490-t003:** Performance comparison among proposed models.

Model	Accuracy	Speed
MoveNetV2 Lightning	Not very good	Good
MoveNetV2 Thunder	Good	Not very good
BlazePose GHUM Lite	Not very good	Good
BlazePose GHUM Full	Good	Not very good
BlazePose GHUM Heavy	Very good	Bad

**Table 4 ijerph-19-05490-t004:** Main methods for collecting depth data.

	Stereo Vision	Structured Light	Time Of Flight
Range	Good	Low	Good
Accuracy	Good	Excelent	Good
Indoor performance	Good	Excelent	Excelent
Outdoor performance	Good	Low	Low
Cost	Medium	Low	Medium

**Table 5 ijerph-19-05490-t005:** Comparison of the main depth devices (prices last updated in March 2022).

	Depth Technology	Depth Range (m)	Depth Resolution	Frame Rate (FPS)	Development Tools	Price (USD)
Microsoft Kinect v1	Structured light	0.4–3	320 × 240, 640 × 480	30	Kinect for Windows SDK v1.8	Discontinued
Microsoft Kinect v2	Time of flight	0.5–4.5	512 × 424	30	Kinect for Windows SDK 2.0	Discontinued
Microsoft Azure Kinect	Time of flight	0.25–5.46	Narrow Mode: 6540 × 576; Wide Mode: 1024 × 1024	30	Microsoft Azure SDK	399
Intel RealSense D415	Active IR Stereo	0.3–10	1920 × 1080	30–90	Intel RealSense SDK 2.0	259
Intel RealSense D435	Active IR Stereo using Global Shutter Sensors	0.105–10	1280 × 720	30–90	Intel RealSense SDK 2.0	299
ASUS® XtionPro Live	Structured light	0.8–3.5	640 × 480	30	Xtion PRO SDK (discontinued)	Discontinued
Stereolabs ZED 2	Neural Stereo Depth Sensing	0.2–20	4416 × 1242	100	ZED SDK	449
OAK-D Pro	Embedded stereo	0.2–35	1280 × 800	120	DepthAI SDK	299
OAK-D-LITE	Embedded stereo	0.2–19.1	640 × 480	200	DepthAI SDK	149
Acusense A1	Structured light	0.2–2	640 × 400, 1280 × 800	44836	Acusense SDK	966
Orbbec Astra (PRO)	Infrared Coded Structured Light	0.6–8	640 × 480	30	Astra SDK	149
ifm O3X100	Time of flight	0.05–3	224 × 172	20	third-party tools	675
e-Con Systems Tara Stereo Camera	Embedded stereo	0.05–0.3	752 × 480	60	third-party tools	299
Nerian Scarlet 3D Depth Camera	Embedded stereo	0.14–to infinity	2432 × 2048	120	third-party tools	not available

**Table 6 ijerph-19-05490-t006:** Comparison of the SDKs mentioned in Table 5 considering their main features and hardware/software requirements.

	SO	Processor	RAM	GPU	Skeleton Tracking	Face Tracking	Price	Support
Kinect for Windows SDK v1.8	Windows 7, Windows 8	Dual-core 2.66-GHz or faster processor	2 GB	-	Yes	Yes	Free	Discontinued
Kinect for Windows SDK 2.0	Windows 8	Physical dual-core 3.1 GHz	4 GB	DX11 capable graphics adapter	Yes	Yes	Free	Discontinued
Microsoft Azure SDK	Windows 10, Ubuntu 18 or later version	Seventh Gen Intel CoreTM i5 Processor (Quad Core 2.4 GHz or faster)	4 GB	NVIDIA GEFORCE GTX 1050 or equivalent	Yes	No	Free	Active
Intel RealSense SDK 2.0	Windows 10, Ubuntu 18.04	6th to 10th generation Intel Core™ and Xeon Processors	-	Intel Iris Pro, Intel HD Graphics 520, 530, 630	Yes	Yes	Free	Active
Xtion PRO SDK (discontinued)	-	-	-	-	-	-	-	Discontinued
ZED SDK	Windows 10, Ubuntu 16.04 or 18.04	Quad-core 2.7 GHz or faster	8 GB	GTX1060 or higher	Yes	third-party tools	Free	Active
DepthAI SDK	Windows 10, Ubuntu, macOS, Raspberry Pi OS, Jestson Nano/Xavier	-	-	-	third-party tools	third-party tools	Free	Active
Acusense SDK	Windows 10, Ubuntu 18.04	-	-	-	third-party tools	third-party tools	Free	Active
Astra SDK	Windows, Linux, Android	x86 processor 1.8 GHz	4 GB	-	third-party tools	third-party tools	Free	New version 2021 in beta (Orbbec SDK Beta))

**Table 7 ijerph-19-05490-t007:** Attribute list.

Attribute	Sub-Attribute	Description	Index
Usefulness		Importance given to the utility derived from its use.	$j_{1}$
Cost of system		Importance given to the cost of the total system considering the cost savings of each alternative.	$j_{2}$
Easiness	Use	Importance given to the ease of use of the system.	$j_{3}$
	Technical Development	Importance given to the ease of technical development of the system.	$j_{4}$
	Maintenance	Importance given to the ease of maintenance of the system.	$j_{5}$
Privacy		Importance given to the use of different devices that may invade privacy.	$j_{6}$

**Table 8 ijerph-19-05490-t008:** Information used for ranking.

Expert	Role	Professional Position	Experience (in Years)	Level of Education	Index
E1	Developer	Professionals, scientists and intellectuals	15–20	Doctorate (Ph.D.)	$i_{1}$
E2	Developer	Professionals, scientists and intellectuals	10–15	Doctorate (Ph.D.)	$i_{2}$
E3	Developer	Technicians and mid-level and professionals	10–15	Upper Secondary	$i_{3}$
E4	Therapist/Physiotherapist	Professionals, scientists and intellectuals	10-15	Ordinary degree	$i_{4}$
E5	Therapist/Physiotherapist	Professionals, scientists and intellectuals	5–10	Ordinary degree	$i_{5}$
E6	Therapist/Physiotherapist	Technicians and mid-level and professionals	5–10	Postgraduate	$i_{6}$
E7	End-user/Caregiver	Elementary occupations	1–5	Lower Secondary	$i_{7}$
E8	End-user/Caregiver	Elementary occupations	more than 20	Primary	$i_{8}$
E9	End-user/Caregiver	Technicians and mid-level and professionals	more than 20	Technical/vocational	$i_{9}$

**Table 9 ijerph-19-05490-t009:** Significance and ranking of the experts.

Experts	Weight	Rank
E1	0.0505	7
E2	0.0442	8
E3	0.0393	9
E4	0.0884	4
E5	0.0707	5
E6	0.0589	6
E7	0.1178	3
E8	0.1767	2
E9	0.3535	1

**Table 10 ijerph-19-05490-t010:** Significance and ranking of the attributes.

Attributes	Weight	Rank
(C1) Usefulness	0.3225	1
(C2) Cost of the system	0.1291	3
(C3) Easiness. Use	0.2457	2
(C4) Easiness. Technical development	0.1159	4
(C5) Easiness. Maintenance	0.1095	5
(C6) Privacy	0.0773	6

**Table 11 ijerph-19-05490-t011:** Significance and ranking of the alternatives.

Alternative	Weight	Rank
(A1) Alternative 1	0.2958	2
(A2) Alternative 2	0.1916	4
(A3) Alternative 3	0.3190	1
(A4) Alternative 4	0.1936	3

## Data Availability

Not applicable.

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
