# Peer review of "Phyx.io: Expert-Based Decision Making for the Selection of At-Home Rehabilitation Solutions for Active and Healthy Aging"

_ijerph, 2022, doi:10.3390/ijerph19095490_

Round 1

Reviewer 1 Report

This article analyzes alternatives for the development of physical activity in adults using different technologies, it is an interesting work and has many implications for an age group that needs to be attended, due to the burden of disease that it represents and by the sum of risk factors, as well as inactivity. Notwithstanding the originality and transcendence of this writing, it presents some considerations that should be valued to increase its impact. Below I indicate my comments.

Indicate in the abstract the objective of the work and present a brief conclusion.

In the introduction, important aspects are mentioned regarding the subject of study, however, it turns out to be very extensive, I recommend narrowing it down, emphasizing those most relevant points of the study, that is, the importance of physical activity at home and the advantages of virtual reality in physical activity.

I recommend deleting the information on lines 117-123 because it is obvious to the reader how the article will unfold.

It is suggested to include the objective of the study at the end of the introduction.

2.2 Please indicate the number of participants, as well as other characteristics that could be relevant to readers, the description given to the authors is very general.

Please fix the tables and place them immediately after citing them in the text (tables 2 and 3 are before being cited, table 5 is in another section, while table 11 is not referenced within the text of the article). Present the tables in an appropriate way so as not to generate confusion in the readers.

Why are figures 5 and 6 in the discussion? It gives the impression that the authors were not careful when presenting their results and graphic resources.

The topic of study is very interesting and has many implications, I recommend the authors to further expand the discussion and focus on other topics that can enrich it, it is striking that they only mention 3 references for a topic that can be discussed from many other dimensions.

The authors need to pay more attention to the presentation of their results (tables and graphs).

Author Response

Response to Reviewer 1 Comments

This article analyzes alternatives for the development of physical activity in adults using different technologies, it is an interesting work and has many implications for an age group that needs to be attended, due to the burden of disease that it represents and by the sum of risk factors, as well as inactivity. Notwithstanding the originality and transcendence of this writing, it presents some considerations that should be valued to increase its impact. Below I indicate my comments.

Point 1: Indicate in the abstract the objective of the work and present a brief conclusion

Response 1:

The abstract section has been rewritten to present the problem, describe the objective of the research and advance the obtained results. The abstract is presented here for the convenience of the reviewer:

While the importance of physical activity in older adults is not in doubt there are important barriers limiting older-adult access to physical exercise. Existing technologies for physical activity in older adults show that despite their positive impacts on health and well-being there is a lack of engagement on exercise due the technology nonacceptance. Usefulness and usability are two major factors for user acceptance along with others such as cost, privacy, required hardware or the maintenance requirements. However, it is unknown the extent to which each factor impacts the user acceptance. Furthermore, other participants, apart from users, should be considered in the decision-making process, such as caregivers and technology providers. Four approaches can be considered for a physical activity support system: a camera-less system, a RGB-camera based system without exercise supervision, an RGB-camera or a depth-camera based system , both with exercise supervision. The main objective of this paper is to propose a Multiple Attribute Decision-Making problem, based on the Ordinal Priority Approach (OPA) method, whose solution ensures the technological acceptance, success, and feasibility of the proposed solution. Results show that the RGB-camera based system with exercise supervision obtains the highest rank.

Point 2: In the introduction, important aspects are mentioned regarding the subject of study, however, it turns out to be very extensive, I recommend narrowing it down, emphasizing those most relevant points of the study, that is, the importance of physical activity at home and the advantages of virtual reality in physical activity.

Response 2:

The introduction has been reviewed and shortened according to the Reviewer suggestions. The introduction highlights the importance of physical exercise, the main barriers found in the literature and the different approaches that have been proposed to support older adults in the performance of physical exercises.

Point 3: I recommend deleting the information on lines 117-123 because it is obvious to the reader how the article will unfold.

Response 3:

Following the reviewer's suggestion the paragraph has been removed.

Point 4: It is suggested to include the objective of the study at the end of the introduction.

Response 4:

This comment is related to the previous point. We have removed the final paragraph of the introductory section and this has been substituted by a paragraph in which the objective of the research is summarised. For the Reviewer convenience, the added paragraph is presented here:

The main objective of this research is therefore the proposal of a Multiple Attribute Decision-Making problem, based on the Ordinal Priority Approach (OPA) method, whose solution ensures the technological acceptance, success, and feasibility of the proposed solution. This paper describes the statement of the mathematical model and the optimization method applied [20] to support the decision on the most appropriate approach for supporting physical activity in older adults.”

Point 5: 2.2 Please indicate the number of participants, as well as other characteristics that could be relevant to readers, the description given to the authors is very general.

Response 5:

In order to present more clearly the information of the participants, the following paragraphs have been added to section 2.2.:

Therefore, the group of participants considered for this study is composed of nine experts representing the different stakeholders of this Pilot Theme, aged between 30 and 68 years old (mean age=45 years old).”

Three developer experts with an average age of 40 years old participated in the study.”

Three therapist or physiotherapist experts with an average age of 35 years old participated in the study.”

Three end-user or caregiver experts with an average age of 61 years old participated in the study.”

In the section 2.4.2., the information of each of the experts who participated in the study is detailed, according to the role, professional position, experience and educational level. All this information will be used to make a ranking of the experts who participated in the study and, in this way, it will be possible to determine the weight in their decisions.”

This last paragraph refers to section 2.4.2. which compiles all the information collected on each of the experts who participated in the study.

Point 6: Please fix the tables and place them immediately after citing them in the text (tables 2 and 3 are before being cited, table 5 is in another section, while table 11 is not referenced within the text of the article). Present the tables in an appropriate way so as not to generate confusion in the readers.

Response 6:

The paper has been edited using Latex and this is the reason why tables and figures were presented in places differently from where they were cited. We have fixed such aspect and we have forced tables and figures to appear where they were being referenced.

Point 7: Why are figures 5 and 6 in the discussion? It gives the impression that the authors were not careful when presenting their results and graphic resources.

Response 7:

This is due to the Latex processor. We have fixed such aspects.

Point 8: The topic of study is very interesting and has many implications, I recommend the authors to further expand the discussion and focus on other topics that can enrich it, it is striking that they only mention 3 references for a topic that can be discussed from many other dimensions.

Response 8:

The discussion has been extended expanding the importance that supervision has on technology acceptance. In this sense, several references have been employed supporting the results obtained in this research. For the convenience of the Reviewer, we present underneath and extract from the discussion section:

Alternative 3 is closely followed by alternative 1. The main difference among these two approaches, in comparison with Alternative 2, is the supervision functionality offered by Alternative 3. The importance that supervision or physical exercise monitoring has on user

engagement to physical therapies has already been proven in the state of the [58 –60]. The obtained result confirms the importance that receiving feedback and tracking evolution has on user acceptance, as stated in [49]. The accuracy of the feedback is not considered more important than the price as it can be concluded from the fact that the Alternative 4 is ranked similarly to Alternative 2, which does not provide any supervision feedback. The improvement in accuracy resulting from the use of a depth camera does not compensate for the increased complexity and price associated therewith.

Point 9: The authors need to pay more attention to the presentation of their results (tables and graphs).

Response 9:

Tables and graphs have been reviewed and corrections have been undertaken for the detected typos and presentation details.

Reviewer 2 Report

  1. I agreed to conduct the review because I was interested in the research problem posed in the title. Underpinning the research is an important report from the European Commission. It is new, dating from 2021. It will probably become a reference for the verification of theoretical and practical assumptions of many proposals in the scientific area under study. However, we know that Member States are responsible for their health strategies and politicians may implement these successful international studies differently. However, it is good that they are there. Unfortunately, there has recently been a war in Ukraine and I read about various cameras being used to kill or defend themselves. I hope that we will soon return to their successful use in medicine.
  2. Certainly the high standard of the article was influenced by the fact that the SHAPES project came from 14 countries. The article fully complies with international scientific standards in terms of the methodology of research presentation.
  3. The introduction contains all the current issues related to the problem of ageing and the challenges for different areas of science. The methodology is well described and based on sound scientific ways of knowing. The definitions are precisely formulated. I cannot evaluate the mathematical model because I am not well versed in it and did not see from the abstract that I would come to evaluate it. Knowing beforehand I would have considered my resignation to do the review, although I am glad to have learned about the research.All further procedures are logical and coherent. In my area of research, I agree with what is synthetically shown in 243-252, that: „Human-body pose estimation, whether involving motion or not, is one of the main challenges addressed by computer vision, and one on which progress continues to be made. Estimating a human-body pose, in 2D, from a monocular image is a task that, to the date, can be successfully achieved. The challenge is found in performing such estimation in 3D, because of the great ambiguity in the 2D positions of a human-body joints estimated from a 2D image. An additional limitation is the computational overhead that is required to calculate these 3D positions, which undeniably impacts on the system performance. There are different alternatives that can be employed to estimate 3D body pose from monocular images. One of the alternatives consists on the use of MoveNet Singlepose model based on the architecture MoveNetV2”. All the procedures described and their tabulation show a clear scientific message.
  4. The only doubt relates to the approach to long-term care facilities. Perhaps I did not fully understand a few sentences. However, it seems to me that long-term care facilities are unfortunately also cutting costs and making savings. „For example, the cost of the system can be a determinant factor for end users, who have to pay for it, but not that important for specialist of long-term care facilities as this will enable to save costs while providing access to exercise support and monitoring to more people” (82-84). I am sending you one sample report showing how often staff are replaced there. https://generations.asaging.org/solving-long-term-care-facility-crisis (Lori Smetanka is the Executive Director of the National Consumer Voice for Quality Long-Term Care in Washington, DC.)

Why am I writing about this? The word 'selecting' in the title troubles me. Perhaps it would be better to consider this word in order to avoid a future debate that technology will replace "redundant" people. It is possible to put it another way. I absolutely do not insist on changing the title. He may not change anything. I just wonder if the authors see this problem in these centres? Although it could technically sound like this, for example: „New technologies to support physical activity at home for older adults”.

  1. I rate the article as original and fully support its printing without amendments. If you find my final question justified I would be grateful for your response - it stems from my interest in the public and journalistic reception of the application of new technologies in various areas of science. However, if you do not find it sufficiently „explained”, please do not bother.

Congratulations on your scientific result!

Author Response

Response to Reviewer 2 Comments

I agreed to conduct the review because I was interested in the research problem posed in the title. Underpinning the research is an important report from the European Commission. It is new, dating from 2021. It will probably become a reference for the verification of theoretical and practical assumptions of many proposals in the scientific area under study. However, we know that Member States are responsible for their health strategies and politicians may implement these successful international studies differently. However, it is good that they are there. Unfortunately, there has recently been a war in Ukraine and I read about various cameras being used to kill or defend themselves. I hope that we will soon return to their successful use in medicine.

Certainly the high standard of the article was influenced by the fact that the SHAPES project came from 14 countries. The article fully complies with international scientific standards in terms of the methodology of research presentation.

The introduction contains all the current issues related to the problem of ageing and the challenges for different areas of science. The methodology is well described and based on sound scientific ways of knowing. The definitions are precisely formulated. I cannot evaluate the mathematical model because I am not well versed in it and did not see from the abstract that I would come to evaluate it. Knowing beforehand I would have considered my resignation to do the review, although I am glad to have learned about the research.All further procedures are logical and coherent. In my area of research, I agree with what is synthetically shown in 243-252, that: „Human-body pose estimation, whether involving motion or not, is one of the main challenges addressed by computer vision, and one on which progress continues to be made. Estimating a human-body pose, in 2D, from a monocular image is a task that, to the date, can be successfully achieved. The challenge is found in performing such estimation in 3D, because of the great ambiguity in the 2D positions of a human-body joints estimated from a 2D image. An additional limitation is the computational overhead that is required to calculate these 3D positions, which undeniably impacts on the system performance. There are different alternatives that can be employed to estimate 3D body pose from monocular images. One of the alternatives consists on the use of MoveNet Singlepose model based on the architecture MoveNetV2”. All the procedures described and their tabulation show a clear scientific message.

The only doubt relates to the approach to long-term care facilities. Perhaps I did not fully understand a few sentences. However, it seems to me that long-term care facilities are unfortunately also cutting costs and making savings. „For example, the cost of the system can be a determinant factor for end users, who have to pay for it, but not that important for specialist of long-term care facilities as this will enable to save costs while providing access to exercise support and monitoring to more people” (82-84). I am sending you one sample report showing how often staff are replaced there. https://generations.asaging.org/solving-long-term-care-facility-crisis (Lori Smetanka is the Executive Director of the National Consumer Voice for Quality Long-Term Care in Washington, DC.)

Why am I writing about this? The word 'selecting' in the title troubles me. Perhaps it would be better to consider this word in order to avoid a future debate that technology will replace "redundant" people. It is possible to put it another way. I absolutely do not insist on changing the title. He may not change anything. I just wonder if the authors see this problem in these centres? Although it could technically sound like this, for example: „New technologies to support physical activity at home for older adults”.

I rate the article as original and fully support its printing without amendments. If you find my final question justified I would be grateful for your response - it stems from my interest in the public and journalistic reception of the application of new technologies in various areas of science. However, if you do not find it sufficiently „explained”, please do not bother.

Response:

We would like to thank Reviewer 2 for the provided feedback. We also thank him/her for the point raised regarding the title with which we totally agree. Based on his/her suggestion, we have proposed the following title that better describes the objective and purpose of the paper: “Phyx.io: An optimal technological approach to at-home physical activity support for older adults

We believe that this title captures the essence of having rigorously evaluated the aspects (attributes) that impacts on technological acceptance. The obtained results therefore suggest that the most appropriate approach is that proposed in Alternative 3 which is the one being implemented in Phyx.io. We can therefore claim that the proposed approach is optimum in terms of the different attributes considered in the evaluation.

Reviewer 3 Report

The manuscript cover a topic of great interest in the sector of e-Health, and is presented in an articulated way, justified and motivated, and expressed in very good English.

It presents a study that tries to propose a solution to the problem of acceptance and adoption, among other questions, in the use of IT tools for aging and rehabilitation and ADLs questions.

One of the critical points identified in this sector (not only by the authors)  is to involve all the actors in the design of the solution for a better adoption and acceptance, we are talking of rehabilitators, ICT designers, end-users, secondary carers, etc.

The problem - as the authors well explain - consists in finding a balance between different opinions, positions and capacities, and to include in the process the non-negligible opinions of the end users which are the final adopters.

The literature presents some alternatives to solve decision making problems, some of them arise  from the field of operational research. Some of these solutions present some obstacles, such as having to give an evaluation of all the attributes involved even if the expert does not actually have skills or experience of them, and this can contribute to generate "artifacts", information that can be  non correctly weighted.

The authors propose a recently introduced method called OPA (Ordinal Priority Approach) to perform a Multiple Attribute Decision-Making process.  The OPA pose the advantage of deriving the weight of opinions and allows the evaluation of the attributes by  ranking them, this consent to not  be forced to specify any evaluation on attributes  that are unknown to the expert.

The method is proposed and  evaluated in its full parts, applied to 4 different alternatives, showing the quality of the method in this specific context (aging and rehabilitation).

The manuscript presents a practical solution based on a general criterion (OPA for MADM) to a very relevant and significant user design problem to support a co-design methodology in an objective way, or in any case based on objectively definable criteria.

There are no exposure problems:  the manuscript is , in my opinion, neat and clear, and well  presented. The work is thorough.

Quite interesting are also the links to the videos given in the footnotes , which  tend to clarify the actual contest of the research and the interviews.

Results are coherent with other works and findings, showing the method is consistent:  the paper is interesting and can be of great utility for the scientific community in this area. Publishing is  very recommended.

-----

I note only a few editing questions in tables 5-6-7-8: a greater spacing between the lines seems necessary to improve the otherwise very complex reading, the lines sometimes get confused with each other.

And the lack of an "of" on line 427

"considering the opinion different experts in the ..."

"considering the opinion * of * different experts in the ..."

Author Response

Response to Reviewer 3 Comments

The manuscript cover a topic of great interest in the sector of e-Health, and is presented in an articulated way, justified and motivated, and expressed in very good English.

It presents a study that tries to propose a solution to the problem of acceptance and adoption, among other questions, in the use of IT tools for aging and rehabilitation and ADLs questions.

One of the critical points identified in this sector (not only by the authors)  is to involve all the actors in the design of the solution for a better adoption and acceptance, we are talking of rehabilitators, ICT designers, end-users, secondary carers, etc.

The problem - as the authors well explain - consists in finding a balance between different opinions, positions and capacities, and to include in the process the non-negligible opinions of the end users which are the final adopters.

The literature presents some alternatives to solve decision making problems, some of them arise  from the field of operational research. Some of these solutions present some obstacles, such as having to give an evaluation of all the attributes involved even if the expert does not actually have skills or experience of them, and this can contribute to generate "artifacts", information that can be  non correctly weighted.

The authors propose a recently introduced method called OPA (Ordinal Priority Approach) to perform a Multiple Attribute Decision-Making process.  The OPA pose the advantage of deriving the weight of opinions and allows the evaluation of the attributes by  ranking them, this consent to not  be forced to specify any evaluation on attributes  that are unknown to the expert.

The method is proposed and  evaluated in its full parts, applied to 4 different alternatives, showing the quality of the method in this specific context (aging and rehabilitation).

The manuscript presents a practical solution based on a general criterion (OPA for MADM) to a very relevant and significant user design problem to support a co-design methodology in an objective way, or in any case based on objectively definable criteria.

There are no exposure problems:  the manuscript is , in my opinion, neat and clear, and well  presented. The work is thorough.

Quite interesting are also the links to the videos given in the footnotes , which  tend to clarify the actual contest of the research and the interviews.

Results are coherent with other works and findings, showing the method is consistent:  the paper is interesting and can be of great utility for the scientific community in this area. Publishing is  very recommended.

Point 1: I note only a few editing questions in tables 5-6-7-8: a greater spacing between the lines seems necessary to improve the otherwise very complex reading, the lines sometimes get confused with each other.

Response 1:

Tables have been amended to increase the interlining space.

Point 2: And the lack of an "of" on line 427

"considering the opinion different experts in the ..."

"considering the opinion * of * different experts in the ..."

Response 2:

We would like to thank the Reviewer for pointing us out the typo. This has been corrected.
